# Research on the SAW Gyroscopic Effect in a Double-Layer Substrate Structure Incorporating Non-Piezoelectric Materials

**DOI:** 10.3390/mi14101834

**Published:** 2023-09-26

**Authors:** Hengbiao Chen, Lili Meng, Mengjiao Lu, Ziwen Song, Wen Wang, Xiuting Shao

**Affiliations:** 1School of Information Science and Engineering, Shandong Normal University, Jinan 250014, China; stcnsuy@163.com (H.C.); mengll_83@hotmail.com (L.M.); 15215419992@163.com (M.L.); sonorouszw@163.com (Z.S.); 2State Key Laboratory of Acoustics, Institute of Acoustics, Chinese Academy of Sciences, Beijing 100190, China; wangwenwq@mail.ioa.ac.cn

**Keywords:** reflection coefficient, SAW angular velocity sensor, SAW gyroscopic effect, double-layer substrate structure, three-dimensional model

## Abstract

The SAW (surface acoustic wave) gyroscopic effect is a key parameter that reflects the sensitivity performance of SAW angular velocity sensors. This study found that adding a layer of non-piezoelectric material with a lower reflection coefficient than that of the upper-layer material below the piezoelectric substrate to form a double-layer structure significantly enhanced the SAW gyroscopic effect, and the smaller the reflection coefficient of the lower-layer material, the stronger the SAW gyroscopic effect, with values being reached that were two to three times those with single-layer substrate structures. This was confirmed using a three-dimensional model, and the experimental results also showed that the thickness of the piezoelectric layer and the type of the lower-layer material also had a significant impact on the SAW gyroscopic effect. This novel discovery will pave the way for the future development of SAW angular velocity sensors.

## 1. Introduction

The SAW angular velocity sensor is a novel type of sensor predicated on the SAW gyroscopic effect [1] that measures rotation angular velocity by detecting the frequency shift induced by rotation [2]. Compared with traditional mechanical and optical sensors, SAW angular velocity sensors possess a series of superior characteristics, such as their simple structure, small size, extended lifespan, and cost effectiveness [3,4,5], while also having significant development potential in the fields of attitude detection and motion control [6,7]. However, SAW devices, including SAW angular velocity sensors, are difficult to apply in reality due to problems such as their weak gyroscopic effect, low frequency, and slow response [8]. To develop SAW devices with high performance, researchers have persistently delved into the exploration of innovative structures and materials, among which layered substrate structures in which piezoelectric materials are combined with non-piezoelectric materials [9] have become an important research direction due to their ability to give full play to the advantages of each material layer [10]. Related studies have shown that layered substrate structures can effectively improve the performance of SAW devices, leading them to be of great reference value for improving the performance of SAW angular velocity sensors.

The performance advantages of layered substrate structures have been demonstrated in a variety of applications. Shen et al. proposed a filter based on the Cu/15°Y-X LiNbO_3_/SiO_2_/Si structure, achieving a bandwidth of 31.08% [11]. Pan et al. developed an SAW pressure sensor with a SiO_2_/Mo/AiN/Mo/Si/SiO_2_ structure that was able to operate in high-temperature and high-pressure environments [12]. Shen et al. employed a SiNx/SiO_2_/IDTs(Cu)/15°Y-X LiNbO_3_ structure, leading to the development of an ultra-small, unpackaged SAW device [13]. Hsu et al. adopted an LiNbO_3_/SiO_2_/Si structure, achieving a high electromechanical coupling coefficient of 35% [14]. Ondo et al. utilized a Pt/AlN/Sapphire structure to engineer an SAW sensor capable of high-frequency operation at temperatures of up to 500 °C [15]. Ma et al. conducted tests on an SAW sensor with a 128°Y–X LiNbO_3_/SiO_2_/Si structure, finding that its performance was superior in all respects [16]. Although SAW devices with layered substrate structures have demonstrated good performance and application prospects in a variety of fields, it is lamentable that there have so far been few reports of the applicastion of layered substrate structures to SAW angular velocity sensors.

Since the SAW is a mechanical wave that propagates along the surface of a piezoelectric substrate or film, it is very sensitive to surface disturbances. Based on the piezoelectric effect, the input interdigital transducer (IDT) can generate an SAW that propagates along the surface of the piezoelectric substrate, which can then be converted back into an electrical signal by the output IDT [17,18]. When the SAW propagates, the Coriolis force generated by the rotation of the SAW angular velocity sensor acts on the vibrating particles [19] and generates a secondary pseudo-SAW, coupled with the original SAW [20], leading to changes in the phase velocity of the SAW, a process known as the SAW gyroscopic effect [21]. On the basis of this, in this paper, the dual delay line angular velocity sensor with distributed metal dot array is taken as the research object, the structure of which is shown in Figure 1, and which is composed of IDTs, metal dot arrays, and a layered substrate. In this sensor, the parallel and reversed SAW dual delay lines form a differential sensing structure to improve the detection sensitivity [22]. Therefore, according to the definition, the optimization of the SAW gyroscopic effect mainly will mainly involve three key components: the substrate, the IDT, and the metal dot array. Methods for the optimization of the IDT have already been described detail in ref. [23]. In terms of the acoustic wave propagation path, setting a metal dot array composed of metal dots with the same size in a periodic arrangement can enhance the Coriolis force, thereby improving the SAW gyroscopic effect. This has been discussed in detail in refs. [24,25,26]. In this paper, we primarily explore methods for increasing the SAW gyroscopic effect, thereby enhancing the sensitivity of sensor detection by means of optimal substrate selection and design.

The layered structure studied in this paper, which includes an upper piezoelectric layer and a lower non-piezoelectric layer, both of which are composed of materials with different reflection coefficients, allows for a high degree of flexibility in optimizing the SAW gyroscopic effect by adjusting the type of the lower-layer material and the thickness of the upper-layer material. The corresponding optimization design method is the focus of this paper. We utilized simulation software to build a three-dimensional model, and in the model, non-piezoelectric materials—Si, SiC, and diamond—with low reflection coefficients, as well as SiO_2_, with a high reflection coefficient, were successively selected for the lower layer of the double-layer substrate for the simulation, with the aim of identifying the correlation between the SAW gyroscopic effect and the reflection coefficient. Furthermore, we analyzed the impact of the type of non-piezoelectric material in the double-layer substrate and the normalized thickness of the piezoelectric layer on the optimal distribution parameters of the metal dot array in the acoustic wave propagation path.

## 2. Theoretical Analysis

The Rayleigh wave, a type of SAW, is an elastic wave that propagates along the surface of a medium [27]. Its energy is confined to one to two wavelengths at the surface of the substrate, and decays exponentially with depth [28]. Concurrently, particles on the surface vibrate in a counterclockwise elliptical trajectory in the plane perpendicular to the surface and parallel to the direction of propagation [29]. The propagation schematic of the Rayleigh wave is depicted in Figure 2.

Related studies have shown that the change in reflection coefficient has an extremely significant impact on the performance of SAW devices [30,31,32]. In [23], it was explicitly pointed out that the reflection coefficient of IDTs has a direct influence on the SAW gyroscopic effect. Inspired by this, our study aims to explore the relationship between the reflection coefficient of the layered substrate and the SAW gyroscopic effect. On the basis of a simulation experiment, we try to determine the optimal design scheme of the layered substrate. Due to the unclear theoretical mechanism by which the reflection coefficient of the layered substrate affects the SAW gyroscope effect, we attempt to analyze it from the perspective of the reflection and transmission of SAW propagation energy. When the Rayleigh wave propagates, a bandgap is generated due to the mass loading effect of the metal dot array distributed on the acoustic wave propagation path, and the characteristic frequency is divided into two [33]: one is the upper stopband boundary frequency $f_{sc+}$, and the other is the frequency of the lower stopband boundary frequency $f_{sc-}$. Considering that Rayleigh wave energy has the characteristic of decaying with depth, in a layered structure, the energy distribution and characteristic frequency of the Rayleigh wave can be changed by adjusting the thickness of different layers within a certain range, thereby affecting the propagation velocity of the Rayleigh wave [31]. For SAW angular velocity sensors, reflection and transmission occur when the acoustic wave energy is transmitted from one layer to another. A part of the energy will be reflected back to the upper medium, affecting Rayleigh wave propagation in the normal direction on the surface. Therefore, the reflection and transmission of energy inside the medium will cause the phase velocity to change in the SAW, thus affecting the SAW gyroscopic effect. The strength of the influence is determined by the reflection and transmission coefficient of the medium. The use of lower-layer materials that possess a lower reflection coefficient than that of the upper-layer material can effectively reduce the energy reflection, thus reducing the interference of the surface Rayleigh wave.

Specifically, in a layered substrate, the lower-layer material having a reflection coefficient that is lower than that of the upper-layer material can reduce the interference and enhance gyroscopic effect. It is well known that, in SAW angular velocity sensors, the intensity of the SAW gyroscopic effect is directly influenced by the coupling strength between o the riginal SAW and the secondary SAW. Simultaneously, the energy propagating into the medium produces a certain reflection, and the energy reflected back to the upper layer interferes with the coupling effect between the original SAW and the secondary SAW, thereby affecting the strength of the SAW gyroscopic effect. At the same time, the lower the reflection coefficient of the lower-layer material, the lower the interference to the Rayleigh wave and the stronger the corresponding gyroscopic effect. According to the relationship between the reflection coefficient and the transmission coefficient R+T=1 [34], in a double-layer substrate structure, when the reflection coefficient of the lower-layer material is low, the corresponding transmission coefficient will be high, so that more energy is transmitted to the lower layer, rather than reflected back to the upper layer, reducing the interference to the surface Rayleigh wave resulting from the reflection of energy in the medium, significantly improving the sensitivity and stability of the SAW angular velocity sensor. Therefore, by adjusting the thickness, reflection coefficient, and transmission coefficient of each layer of the double-layer substrate, the propagation behavior of the Rayleigh wave in the substrate can effectively be controlled, thereby optimizing the performance of the SAW angular velocity sensor. Referring to the properties of Rayleigh wave [35], in order to ensure that the lower-layer material can more effectively affect the energy distribution of the Rayleigh wave in the medium, the thickness of the upper piezoelectric material should be maintained within one wavelength in subsequent modeling.
(1)$$R={\left[\frac{Z_{2}-Z_{1}}{Z_{2}+Z_{1}}\right]}^{2},$$
(2)$$T=\frac{4Z_{1}Z_{2}}{{\left(Z_{2}+Z_{1}\right)}^{2}},$$
where $Z_{1}$ and $Z_{2}$, respectively, represent the characteristic impedances of the upper piezoelectric material and the lower non-piezoelectric material in the double-layer substrate structure. For single-layer piezoelectric materials, $Z_{1}$ and $Z_{2}$ represent the characteristic impedance of the air and the piezoelectric material, respectively. The characteristic impedance can be represented as the product of the material density and acoustic wave velocity [36]:(3)Z=ρV,
where ρ and V, respectively, represent the density of the medium and the velocity of the Rayleigh wave. According to the characteristics of Rayleigh wave propagation on the surface of the substrate [37], in the double-layer substrate structure, when the upper piezoelectric material is determined, if non-piezoelectric material with a high characteristic impedance is selected for the lower substrate, the transmission performance at the interface between the upper and lower materials can be effectively enhanced. This implies that more energy will be transmitted into the lower medium, allowing the acoustic wave to propagate more stably on the surface of the medium, achieving a stronger SAW gyroscopic effect.

## 3. Modeling

In this study, the finite element method (FEM) was used for three-dimensional simulation validation, as it is able to demonstrate various complex vibrations of particles on surfaces, geometric shapes, and interfaces, making it possible to simulate different types of acoustic wave [28]. According to the study in ref. [38], there is a direct relationship between the gain factor g of the SAW gyroscopic effect and the Poisson’s ratio μ of the substrate material, the value of which typically ranges from 0 to 0.5 [39]. Generally, the value of g shows a downward trend as the value of μ increases. Therefore, to enhance the SAW gyroscopic effect, we selected STX quartz, which has a smaller Poisson’s ratio, as the piezoelectric material. Ref. [2] also reported that STX quartz has a strong SAW gyroscopic effect. For the comparison and simulation experiment, SiO_2_, Si, SiC, and diamond were chosen as non-piezoelectric materials to construct double-layer substrate structures. According to Formula (3), the order of these materials can also be derived in order from low to high characteristic impedance: SiO_2_, STX quartz, Si, SiC, diamond.

According to the characteristic whereby the Rayleigh wave can only propagate on the surface of the medium, and its energy is constrained to within 1–2λ and decays rapidly with depth [28], in the double-layer substrate structure, we set the thickness of the piezoelectric layer to 0.6λ to ensure that the acoustic wave could penetrate normally to the lower layer. Additionally, the non-piezoelectric layer was set to 2λ, and a Perfectly Matched Layer (PML) with a thickness of 1λ was set at the bottom to absorb SAW and prevent reflection from affecting the simulation results [40]. Although the distribution of metal dot arrays along the acoustic wave propagation path has been shown to effectively enhance the Coriolis force [25,41], in SAW angular velocity sensors, the size of the metal dot array in terms of length, width, and thickness changes the proportions of the elastic wave energy in the metal dot array, as roughly shown in Figure 3, and the optimal size of the metal dot array varies with the material [26]. Previous research has shown that with increased thickness of the metal dot array, the SAW gyroscopic effect becomes stronger [21], but in this study, we chose a metal dot thickness of 0.9 um due to technological limitations. One metal dot was set within a period, located at the center of the double-layer substrate surface. The length and width of the metal dot array was increased from 1/16λ to 15/16λ in order to find the optimal size for enhancing the SAW gyroscopic effect. The system default ultra-fine setting was used for the grid.

The boundary conditions for the layered model are as follows [42]:(1)Mechanical boundary conditions: The upper surface of the piezoelectric substrate is free, and the bottom layer of the PML is fixed;(2)Electrical boundary conditions: The upper surface of the piezoelectric substrate has zero charge, and the bottom surface of the PML has zero charge or is grounded;(3)The front and back sides and the left and right sides have periodic boundary conditions;(4)The piezoelectric/non-piezoelectric interface has continuous mechanical boundary conditions. Figure 4 provides a three-dimensional depiction of this model.

In this simulation experiment, the upper piezoelectric material is fixed as STX quartz, while the lower-layer materials are alternately chosen from among four materials, including three non-piezoelectric materials—Si, SiC, and diamond—with characteristic impedances greater than that of the upper material, and SiO_2_, a non-piezoelectric material with a characteristic impedance lower than that of the upper material. We aimed to validate the impact of the reflection coefficient of the non-piezoelectric material and the thickness of the piezoelectric layer in the double-layer substrate structure on the SAW gyroscopic effect, as well as the influence on the distribution parameters of the metal dot array. We further verified the experimental results by changing the upper piezoelectric material.

The SAW gyroscopic effect can be obtained by analyzing the characteristic frequency. When a rotation vector is applied to the model, the characteristic frequency of the upper and lower boundaries of the stopband will change accordingly, and their changes will be different, corresponding to the variation in the characteristic frequencies of SAW propagating along the +*x*- and −*x*-axis directions [23]. Therefore, the strength can be represented by the gyroscopic effect’s gain factor g [16].
(4)$$g=\frac{2\pi [(f_{sc+}^{\prime }-f_{sc-}^{\prime })-(f_{sc+}-f_{sc-})]}{\Omega },$$
where Ω represents the introduced rotation vector, $f_{sc+}^{\prime }$ and $f_{sc-}^{\prime }$, respectively, represent the upper boundary frequency and lower boundary frequency of the bandgap after the addition of the rotation vector. The material properties involved in this article are given in Table 1 and Table 2, below, all of which are derived from refs. [30,31,36,43]. At the same time, all of the simulation experiments performed in this article were carried out under the same rotation vector $\Omega =2\times 10^{3}\pi (\mathrm{rad}/s)$, so in the subsequent text, the difference in frequency before and after rotation, $g^{\prime }$, is used to represent the magnitude of the SAW gyroscopic effect
(5)$$g^{\prime }=(f_{sc+}^{\prime }-f_{sc-}^{\prime })-(f_{sc+}-f_{sc-}).$$

## 4. Analysis of the Simulation Results

### 4.1. SAW Characteristics of the Double-Layer Piezoelectric Substrate

Figure 5a displays the variation in SAW velocity V with changes in the normalized thickness $h_{\mathrm{STX}\;\mathrm{quartz}}/\lambda$ of the piezoelectric layer in the single-layer STX quartz substrate and the double-layer substrate combining different lower-layer materials, respectively, where the SAW velocity in the single-layer STX quartz substrate is represented by $V_{q}$. Figure 5b illustrates the effect of changes in the normalized thickness $h_{\mathrm{STX}\;\mathrm{quartz}}/\lambda$ of the piezoelectric layer on the difference in SAW velocity ΔV before and after rotation in a double-layer substrate structure with Si, SiC, and diamond as the lower-layer materials, respectively, where ΔV is an effective measure for the SAW gyroscopic effect. It should be pointed out that no metal dot array was added in either Figure 5a or Figure 5b. In Figure 5a, with the increase in $h_{\mathrm{STX}\;\mathrm{quartz}}/\lambda$, each curve initially shows a trend of rapid decline, which then slows down and gradually approaches $V_{q}$. The reason for this phenomenon is that when the thickness of the piezoelectric layer decreases, the fluctuations in the Rayleigh wave penetrate the piezoelectric layer directly, and are affected by the non-piezoelectric layer. Conversely, when the thickness increases, the fluctuations are confined to the piezoelectric layer, which indicates that the thickness of the piezoelectric layer directly affects the velocity of the Rayleigh wave, which is consistent with the conclusions drawn in ref. [44]. The SAW velocity V for different types of double-layer substrate strucutre show significant differences. When the lower-layer material is SiO_2_, the V curve almost coincides with $V_{q}$, and the velocity difference is only about 52 m/s. However, when the lower-layer materials are Si, SiC, and diamond, the velocity differences reach 911 m/s, 2739 m/s, and 2869 m/s, respectively. This also suggests that in a double-layer substrate structure, the greater the difference in reflection coefficients between the upper and lower-layer materials, the higher the sensitivity of V. In Figure 5b, we analyze a segment with $h_{\mathrm{STX}\;\mathrm{quartz}}/\lambda$ from 0.6 to 0.8, where the rotational angular velocity is 0.1. From the perspective of vertical comparison, ΔV increases with the increasing difference in the reflection coefficient between the upper- and lower-layer materials. From the perspective of horizontal comparison, ΔV decreases as $h_{\mathrm{STX}\;\mathrm{quartz}}/\lambda$ increases. This indicates that in a double-layer substrate structure, a larger difference in reflection coefficient between the upper and lower materials, as well as a thinner piezoelectric layer, is more beneficial in enhancing the SAW gyroscopic effect.

In Figure 6, the simulation images of Rayleigh wave propagation in the double-layer substrate structure constructed of STX quartz and four different non-piezoelectric materials are presented. In the simulation, the material from which the metal dot array is constructed is Cu, its length and width are uniformly set to 1/16λ, and the thickness is uniformly set to 0.9 um. The lower-layer materials are SiO_2_, Si, SiC, and diamond, respectively, arranged in ascending order of acoustic impedance. In combination with Equation (1), it can be concluded that the reflection coefficient of the lower non-piezoelectric material corresponding to Figure 6a compared to Figure 6d decreases sequentially. At the same time, it can be observed that, as the reflection coefficient decreases, the energy of the Rayleigh wave is concentrated on the surface of the piezoelectric material, and the impact on the part below the surface of the piezoelectric material and on the non-piezoelectric material gradually decreases, even becoming almost invisible. This means that the propagation state of the Rayleigh wave becomes clearer. This result further indicates that when the lower layer is a material with a low reflection coefficient, the energy of the acoustic wave is transmitted to the lower layer to a greater degree than that to which it is reflected back to the upper layer, thereby reducing the impact on the SAW propagation on the surface of the double-layer substrate.

### 4.2. The Influence of the Reflection Coefficient on the SAW Gyroscopic Effect

In Figure 7, the SAW gyroscopic effect in a single-layer STX quartz piezoelectric substrate is compared with that in a double-layer substrate structure combining STX quartz and different non-piezoelectric materials. According to the simulation results, it can be concluded that in the double-layer substrate structure, the reflection coefficient of the lower-layer non-piezoelectric material has a significant impact on the SAW gyroscopic effect. By analyzing the SAW properties when Si, SiC and diamond are selected as the lower-layer materials, in turn, it was found that the SAW gyroscopic effect is more significant in the double-layer substrate where the lower layer consisted of a non-piezoelectric material with a low reflection coefficient, which is consistent with the above analysis. As shown in Figure 7, when the non-piezoelectric materials in the lower layer are Si, SiC, and diamond, when the reflection coefficient is gradually decreased, the maximum SAW gyroscopic effect in the double-layer substrate reaches 50.31 kHz, 54.55 kHz, and 85.4 kHz, respectively, which is 1.7 times, 1.85 times, and 2.89 times the maximum SAW gyroscopic effect in the single-layer STX quartz piezoelectric substrate. This indicates that choosing a non-piezoelectric material with a low reflection coefficient as the lower-layer material effectively enhances the SAW gyroscopic effect.

Additionally, in Figure 7, compared to the single-layer STX quartz piezoelectric substrate, the gyroscopic effect of the STX quartz/SiO_2_ double-layer substrate structure is slightly lower. Since the reflectivity of SiO_2_ is greater than that of STX quartz, it can be concluded that if the reflection coefficient of the lower-layer material is greater than that of the upper-layer material, the energy propagated into the substrate will be more significantly reflected back into the upper layer, causing interference with the Rayleigh wave propagating on the substrate surface and weakening the SAW gyroscopic effect. In our previous research, we found that different substrate structures have different requirements in terms of the optimal size of the metal dot array [26]. Moreover, during the simulation, we observed another phenomenon, whereby the addition of non-piezoelectric materials below the piezoelectric material could also change the optimal size of the metal dot array. For example, in Figure 7, when the lower-layer material is diamond, the length and width of the optimal metal dot array changes from 3/16λ for the single-layer STX quartz substrate structure to 12/16λ. In order to further verify the conclusion that non-piezoelectric material may affect the optimal size of the metal dot array, we conducted a simulation experiment utilizing a 128°YX-LiNbO_3_/SiC double-layer substrate structure, and obtained similar results to those described above, as shown in Figure 8, below.

Figure 8 shows the relationship between the strength of the SAW gyroscopic effect and the length and width of the metal dot array in both the 128°YX-LiNbO_3_/SiC double-layer substrate structure and the 128°YX-LiNbO_3_ single-layer piezoelectric substrate structure. According to the results shown in Figure 8, a stronger SAW gyroscopic effect was observed when SiC with a reflection coefficient lower than that of 128°YX-LiNbO_3_ was used as the lower-layer non-piezoelectric material, and its intensity was 1.43 times higher than that of the single-layer 128°YX-LiNbO_3_ piezoelectric substrate. At the same time, we also observed that the optimal length and width of the metal dot array in 128°YX-LiNbO_3_/SiC also changed, from 5/16λ for the single-layer 128°YX-LiNbO_3_ substrate to 1/16λ. Therefore, we tentatively speculate that the reflection coefficient of the lower-layer material of the piezoelectric substrate may also have an impact on the optimal distribution parameters of the metal dot array on the SAW propagation path.

### 4.3. The Influence of Piezoelectric Layer Thickness on the SAW Gyroscopic Effect

Firstly, the relationship between the SAW gyroscopic effect and the normalized thickness of the piezoelectric layer in 128°YX-LiNbO_3_/SiC and STX quartz/Si substrate structures is simulated, and the results are shown in Figure 9. Under the condition in which metal dots with sizes of 1/16λ and 3/16λ are distributed on STX quartz/Si and 128°YX-LiNbO_3_/SiC substrates, respectively, when the piezoelectric layer thicknesses of both are increased from 0.6λ to 0.9λ, from the figure, a phenomenon can be observed in which, in the STX quartz/Si double-layer substrate, the SAW gyroscopic effect decreases monotonically from 50.31 kHz to 37.3 kHz with increasing thickness of the piezoelectric layer. However, the 128°YX-LiNbO_3_/SiC substrate shows a different situation, with the SAW gyroscopic effect first decreasing and then increasing. When the normalized thickness of the piezoelectric layer is 0.76λ, the strength of the SAW gyroscopic effect drops to its lowest value, 4.4 kHz, and then recovers to the value of the single-layer 128°YX-LiNbO_3_ matched with a metal dot array size of 1/16λ. This result is consistent with the results reported in [16]. Based on the above research results, we further verified the double-layer substrate based on the STX quartz piezoelectric layer, and the specific results are shown in Figure 10.

Figure 10 shows the variation in the SAW gyroscopic effect with different piezoelectric layer thicknesses in STX quartz/SiC and STX quartz/Si substrates distributed with the same metal dot array, with a size of 3/16λ. From the figure, it can be observed that as the thickness of the piezoelectric layer increases in these two double-layer substrate structures, the SAW gyroscopic effect shows a similar trend to that in Figure 9. At the same time, the SAW gyroscopic effects of these two double-layer substrates decrease to their minimum values of 23.34 kHz and 20.57 kHz, respectively, when the thickness of the piezoelectric layer is 2.2λ. Then, the SAW gyroscopic effect starts to increase, approaching the SAW gyroscopic effect observed for the single-layer STX quartz substrate distributed with a metal dot array size of 3/16λ. This is because, as the thickness of the piezoelectric layer gradually increases, the Rayleigh wave decays before propagating to the non-piezoelectric layer. Therefore, the non-piezoelectric layer cannot affect the SAW gyroscopic effect.

Figure 11 illustrates the relationship between the SAW gyroscopic effect and the length and width of the metal dot array in the STX quartz/SiC double-layer substrate as the normalized thickness of the piezoelectric layer is gradually decreased. We found that with the continually decreasing thickness of the piezoelectric layer, the SAW gyroscopic effect was further improved, and its intensity increased from 54.55 kHz to 67.1 kHz. When the normalized thickness of the piezoelectric layer decreased by 0.1λ, the rate of change of the SAW gyroscopic effect reached 23%. At the same time, we also found that when the normalized thickness of the piezoelectric layer changed, the optimal size of the metal dot array matching the double-layer substrate structure also changed. For example, when the normalized thickness of the piezoelectric layer was 0.55λ or 0.5λ, the optimal length and width of the metal dot array were 1/16λ instead of 3/16λ. Therefore, when designing SAW angular velocity sensors, the influence of the material and thickness of each layer of the substrate on the optimal size of the metal dot array distributed along the SAW propagation path should be comprehensively considered.

## 5. Conclusions

This paper draws on relevant experience in the study of the reflection coefficient in SAW devices, and discusses the SAW gyroscopic effect in SAW angular velocity sensors from the perspective of energy reflection inside the substrate. It was found that in double-layer substrate structures, choosing a non-piezoelectric material with a lower reflection coefficient than the upper-layer material as the lower-layer material can effectively enhance the SAW gyroscopic effect, and the lower the reflection coefficient, the better the enhancement effect on the gyroscopic effect. These findings were validated through simulation experiments, where it was shown that lower-layer materials with lower reflection coefficients can reduce the interference to the SAW. In addition, the simulation results also showed that the optimal size of the metal dot array distributed on the double-layer substrate was closely related to the thickness of the piezoelectric layer, as well as the type of the lower-layer material and its reflection coefficient, and the SAW velocity was also intimately related to the difference between the reflection coefficients of the upper- and lower-layer materials. Experiments on the relationship between the normalized thickness of the piezoelectric layer and the SAW gyroscopic effect also showed that reducing the thickness of the upper piezoelectric material in the double-layer substrate further reduces the energy reflection inside the substrate and enhances the SAW gyroscopic effect. Therefore, when designing an SAW angular velocity sensor, it is necessary to comprehensively consider various factors, including the type, materials, and thickness of the substrate, as well as the distribution parameters of the metal dot array, in order to optimize the SAW gyroscopic effect.

## Figures and Tables

**Figure 1 micromachines-14-01834-f001:**
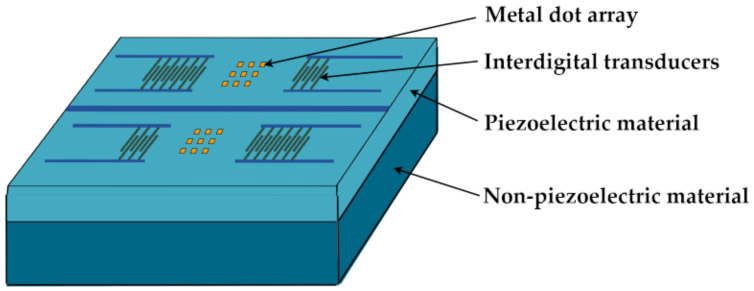
SAW angular velocity sensor structure.

**Figure 2 micromachines-14-01834-f002:**
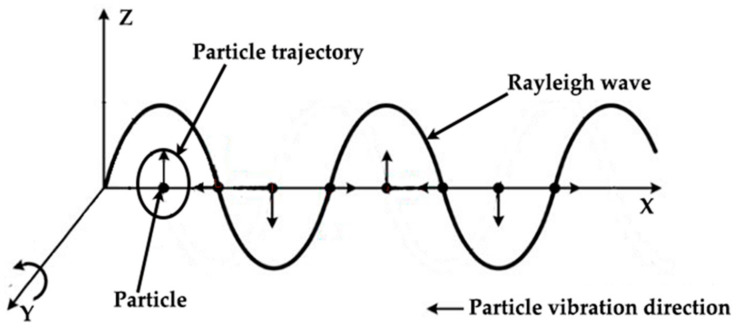
Trajectory of Rayleigh wave propagation on the surface of a medium.

**Figure 3 micromachines-14-01834-f003:**
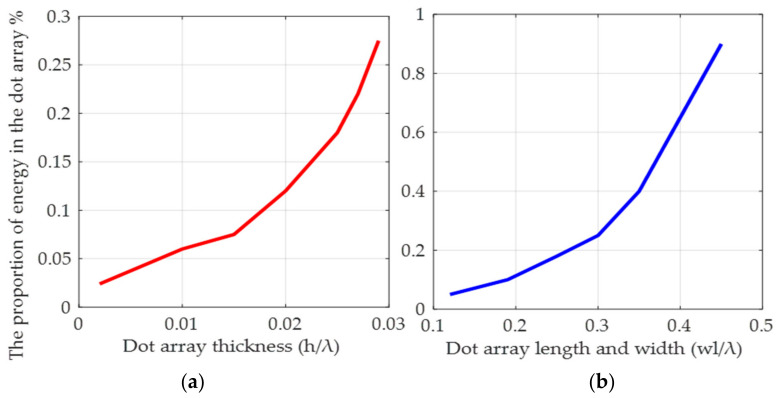
The relationship between the energy proportion in the metal dot array and its size: (**a**) relationship between energy proportion and thickness; (**b**) relationship between energy proportion and length and width.

**Figure 4 micromachines-14-01834-f004:**
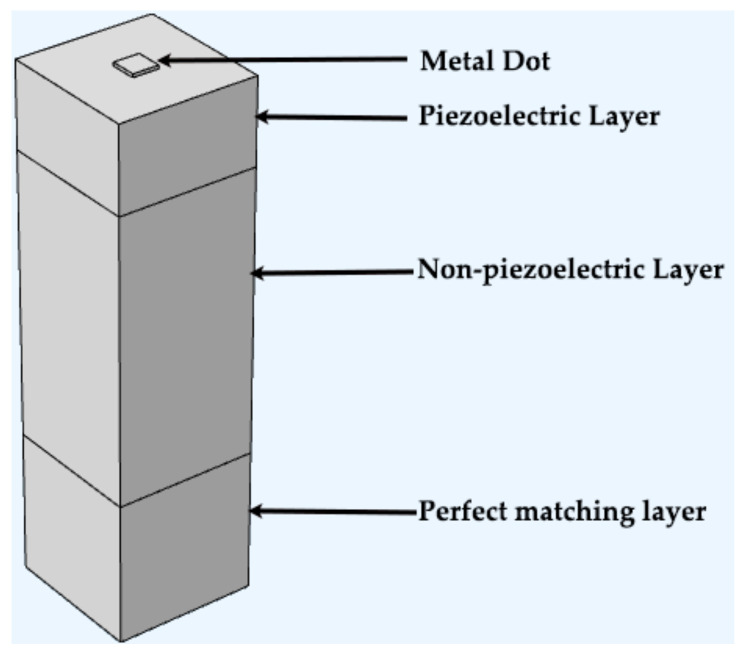
Three-dimensional periodic model in FEM.

**Figure 5 micromachines-14-01834-f005:**
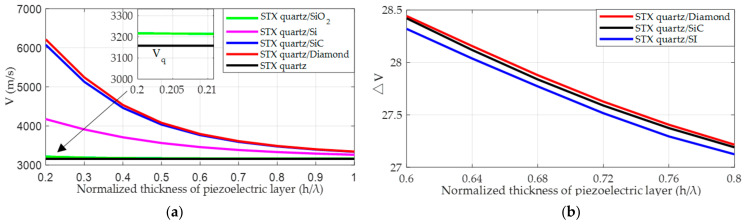
(**a**) Variation in SAW velocity with changes in the normalized thickness of the piezoelectric layer in the double-layer substrate when $h_{\mathrm{STX}\;\mathrm{quartz}}/\lambda$ is between 0.2 and 1; (**b**) variation in the difference in SAW velocity before and after rotation with changes in the normalized thickness of the piezoelectric layer in the double-layer substrate when $h_{\mathrm{STX}\;\mathrm{quartz}}/\lambda$ is between 0.6 and 0.8.

**Figure 6 micromachines-14-01834-f006:**
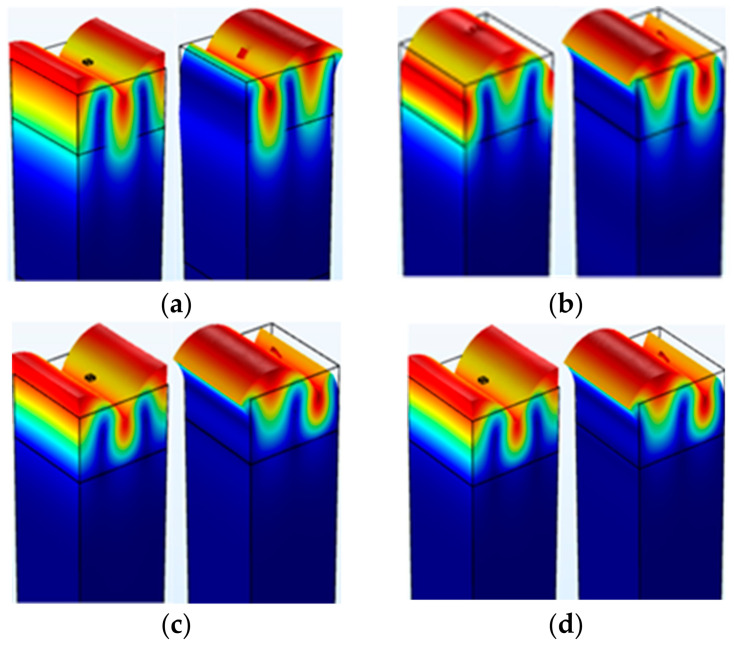
The upper and lower boundary frequency of SAW in double-layer substrate combined with different non-piezoelectric materials: (**a**) SiO_2_; (**b**) Si; (**c**) SiC; (**d**) Diamond.

**Figure 7 micromachines-14-01834-f007:**
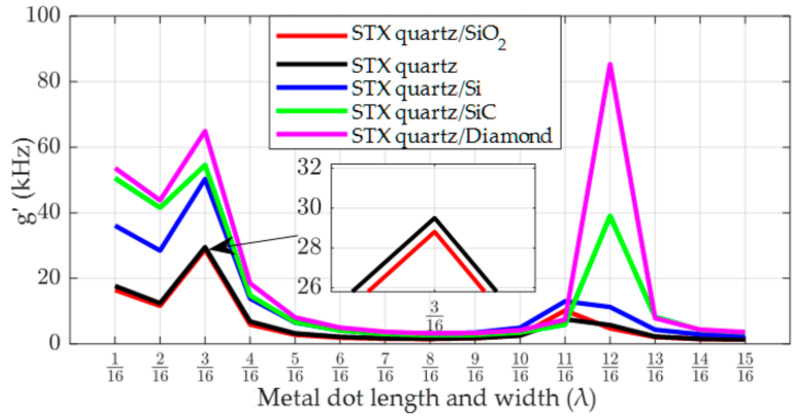
SAW gyroscopic effect in the STX quartz substrate and in a double-layer substrate in which STX quartz is combined with different non-piezoelectric materials.

**Figure 8 micromachines-14-01834-f008:**
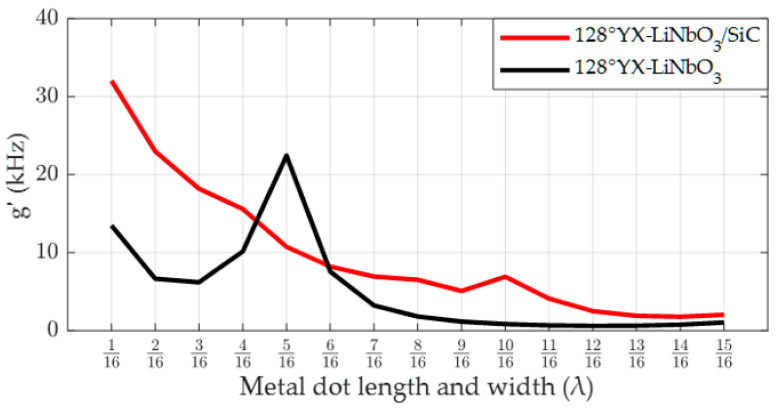
The SAW gyroscopic effect in the 128°YX-LiNbO_3_/SiC substrate versus in the 128°YX-LiNbO_3_ substrate.

**Figure 9 micromachines-14-01834-f009:**
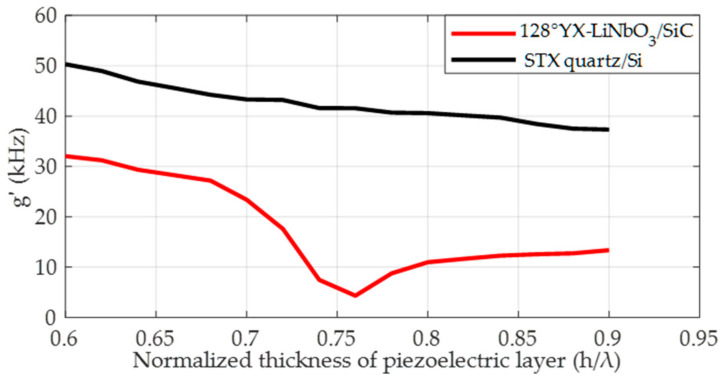
Correlation between normalized thickness of the piezoelectric layer and the AW gyroscopic effect in 128°YX-LiNbO_3_/SiC and STX quartz/Si structures.

**Figure 10 micromachines-14-01834-f010:**
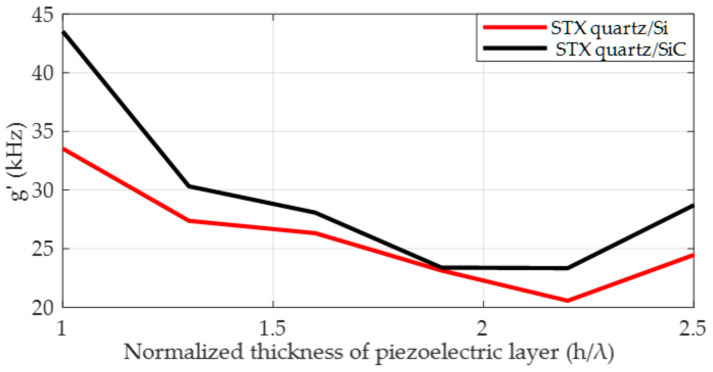
Variation law of the SAW gyroscopic effect with a continuous increase in the normalized thickness of the piezoelectric layer in STX quartz/SiC and STX quartz/Si structures.

**Figure 11 micromachines-14-01834-f011:**
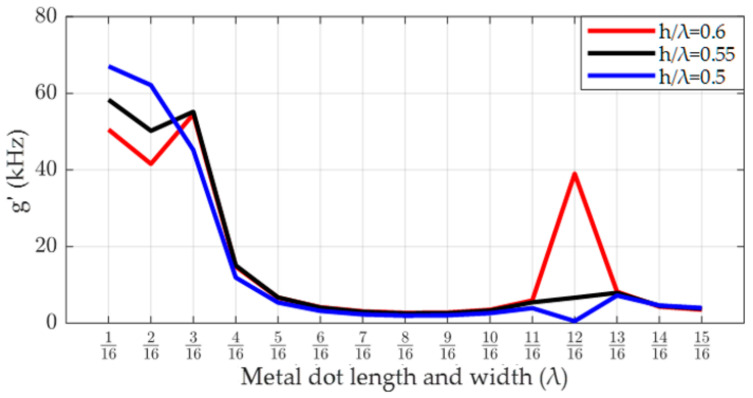
Relationship between the thickness of the piezoelectric layer and the metal dot array distribution parameters in STX quartz/SiC substrate.

**Table 1 micromachines-14-01834-t001:** Material parameters.

Materials	STX Quartz	128°YX-LiNbO_3_	SiO_2_	Si	SiC	Diamond
Density (kg/m^3^)	2651	4628	2200	2329	3216	3515
Young’s Modulus (10^9^ Pa)	72	170	70	170	748	1050
Poisson’s Ratio	0.17	0.25	0.17	0.28	0.45	0.1

**Table 2 micromachines-14-01834-t002:** Constants related to the piezoelectric materials used in the model.

Materials		STX Quartz	128°YX-LiNbO_3_
Stiffness constants (10^11^ N/m^2^)	$C_{11}$	0.87	1.98
	$C_{12}$	0.07	0.54
	$C_{13}$	0.12	0.65
	$C_{14}$	−0.18	0.07
	$C_{33}$	1.07	2.27
	$C_{44}$	0.58	0.59
Piezoelectric constants (C/m^2^)	$e_{x1}$	0.171	
	$e_{x4}$	−0.0436	
	$e_{x5}$		3.69
	$e_{y2}$		2.42
	$e_{z1}$		0.3
	$e_{z3}$		1.77
	$e_{z6}$	0.14	
Dielectric constants (10^−12^ F/m)	$\varepsilon_{11}$	$4.5\times \varepsilon_{0}$	$45.6\times \varepsilon_{0}$
	$\varepsilon_{33}$	$4.6\times \varepsilon_{0}$	$26.3\times \varepsilon_{0}$
	$\varepsilon_{0}$	8.854	8.854

## Data Availability

Not applicable.

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
