# Peer review of "Research on the SAW Gyroscopic Effect in a Double-Layer Substrate Structure Incorporating Non-Piezoelectric Materials"

_micromachines, 2023, doi:10.3390/mi14101834_

Round 1

Reviewer 1 Report

In the paper the SAW angular velocity sensor is considered  on the base of SAW gyroscopic effect which measures rotation angular velocity by detecting the frequency shift induced by rotation. It is a novel type of SAW angular velocity sensing method which is an original solution in the field based on SAW gyroscopic effect.

The simulation results made for the double layer structures with various materials can improve the construction of the real sensors. The performed simulations are analyzed in a proper way and give an interesting remarks for the real constructions.

Interesting subject, however only some simulations are presented. The simulation results should be confirmed by the real measurements and construction of the sensor structures. An additional figure of the sensor structure in the beginning of the paper would improve the understanding of the main conclusions. For better understanding i would suggest to add in the beginnig the figure of the layered structure utilized in the simulations.

it is OK, only too often using the word "effect" sometimes in the same sentence.

Reviewer 2 Report

The article is devoted to the study of the SAW gyroscopic effect in bilayer structures including the non-piezoelectric layer. The increase of gyroscopic effect in bilayer structures in relation to single-layer piezoelectric structures is demonstrated, which is important for creation of sensors of gyroscopic effect. The article is of scientific interest and corresponds to the subject of the journal.

However, a number of serious observations should also be noted:

(1) It is necessary to describe separately in detail the scheme of the investigated sample with an array of metallic dots. It is necessary to present the scheme of the surface of the investigated sample with metal dots. On the one hand, the SAW diffracts on the metal dots, which act as a phonon crystal, and on the other hand, the SAW velocity under the metal dot is smaller than the velocity on the free surface. Accordingly, the configuration of the metal dots on the surface and their size with respect to the acoustic beam can lead to both increased diffraction in the acoustic beam and decreased diffraction in the acoustic beam. Also important is the metal used and its thickness, which affects the surface loading of the piezoelectric substrate.

(2) The article does not address the issue of the change of SAW velocity in a layered structure with respect to the velocity in a single-crystal substrate. For example, see "Dmitry Roshchupkin, Evgenii Emelin, Olga Plotitcina, Anatoly Mololkin and Oleg Telminov, Scanning Electron Microscopy Investigation of Surface Acoustic Wave Propagation in a 41° YX-Cut of a LiNbO3 Crystal/Si Layered Structure, Crystals 2021, 11, 1082 (1-11). https://doi.org/10.3390/cryst11091082", where the increase of SAW velocity in the layered structure is demonstrated. Moreover, the SAW velocity will depend on the thickness of the piezoelectric layer. And for the cases of SiC, Si, and Diamond, the SAW velocities in the top piezoelectric layer will be significantly different due to the difference of acoustic wave velocities in the non-piezoelectric layers.

There are also a number of technical errors in the article that need to be corrected:

(1) formulas (1) and (2) must be followed by commas

(2) line 140: with a small letter and without indentation

(3) formula (3) must be followed by a comma.

(4) line 140: with a small letter and without indentation

(5) lines 188-190: deselect the text

(6) formula (4) must be followed by a comma

(7) line 224: remove the point

(8) formula (5) must be followed by a point

(9) lines 226-227: deselect the text

(10) lines 286-287: deselect the text

(11) line 337: leave one point at the end of the sentence

The English language of the article is correct.

Round 2

Reviewer 2 Report

The authors made a change to the article that significantly increased the scientific component. The article can be published in this revision.

The article is presented using good English language.